# DualTune: Decoupled Fine-Tuning for On-Device Agentic Systems

## ABSTRACT

The deployment of Large Language Models (LLMs) as agentic orchestrators has revolutionized task automation, but the need for privacy-preserving, cost-effective solutions demands on-device inference capabilities. However, local LLMs consistently underperform compared to frontier models in tool calling scenarios, struggling with both tool selection from large tool sets and accurate argument generation for complex parameter structures. We introduce a methodology that disaggregates a tool-calling task into two distinct subtasks: tool selection and argument generation. We propose *decoupled fine-tuning*, a novel post-training approach that employs LoRA fine-tuning to create dedicated LoRA adapters for tool selection and tool-specific argument generation using separate loss masking for each of the subtasks. Furthermore, we present DualTune, an inference framework that leverages the LoRA adapters created using decoupled fine-tuning to perform efficient agent orchestration with the help of local models on end-user devices. DualTune decomposes the tool-call generation step into tool selection and argument generation, and dynamically loads the corresponding LoRA adapters to generate tool calls. Additionally, DualTune implements hierarchical orchestration to restrict the number of tools required for tool selection. Our experiments on the MCP-Bench benchmark demonstrate that the Qwen-2.5-7B model trained using decoupled fine-tuning improves the tool calling accuracy of the base model by 46%, and outperforms other local reasoning, non-reasoning and fine-tuned models of similar size in all cases, and models that are $2\times$ larger, in most cases.

## 1 INTRODUCTION

The emergence of Large Language Models (LLMs) has enabled the development of sophisticated agent systems that can interpret natural language commands and coordinate multiple tools to execute complex, end-to-end tasks. These agent systems use an LLM as the central controller (also called as orchestrator) that receives user requests and a list of available tools. The orchestrator then works through the task step-by-step: it calls a tool, stores the results, uses those results to decide the next action, calls another tool, and builds up a collection of information from each step. The process continues until the controller has gathered enough data to provide a complete response to the user's original request.

The proliferation of agentic architectures has driven the establishment of standardized integration protocols, most notably the Model Context Protocol (MCP) (Anthropic, 2024). MCP establishes a standardized protocol for LLMs to communicate with external applications. Every MCP application defines a set of tools (called as a "toolset" or MCP Server), that contains all the tools belonging to the application, along with their descriptions and instructions on how the LLM can use them. Leveraging this protocol, practitioners have built a vibrant ecosystem, from reading and writing local files to invoking remote APIs (Model Context Protocol (MCP) contributors, 2025), all of which are accessible directly through LLMs.

Nevertheless, incorporating applications into agentic frameworks raises security concerns, as it can involve exposing sensitive or private information residing within these applications to LLMs (Hasan et al., 2025; Hou et al., 2025; Guo et al., 2025). This makes on-device inference critical for agentic systems. Running LLMs locally on end-user devices enhances privacy by ensuring that personal data remains on-device, while simultaneously eliminating the costly API expenses associated with orchestration via frontier models (Gu et al., 2025). However, this shift toward local deployments raises a critical question: Can locally deployed LLMs match the orchestration performance of their cloud-based counterparts?

Our investigation using MCP evaluation benchmarks (Wang et al., 2025) reveals two core short-comings that make existing local LLMs ineffective as agentic orchestrators. First, these models demonstrate **poor tool selection capabilities**, frequently choosing inappropriate tools due to limited ability to reason about the right tool usage from sometimes ambiguous descriptions. This challenge intensifies with larger tool sets, as expanded context length (often exceeding tens of thousands of tokens) overwhelms local models' attention mechanisms. Second, local LLMs exhibit **poor argument generation capabilities**, frequently failing to produce accurate parameters for tools with structural requirements. They also have limited ability to fix the mistakes in subsequent steps, causing repeated failures in tool calling. Furthermore, our experiments show that straightforward approaches such as prompt tuning yield marginal accuracy improvements, but fail to improve local models' orchestration capabilities.

A popular approach for improving the tool-calling capability of local models is fine-tuning (Erdogan et al., 2024; Lin et al., 2024; Liu et al., 2024b). While traditional fine-tuning helps in improving the accuracy of local models for specialized tasks, its improvements in tool-calling performance are not substantial. This is because traditional fine-tuning requires that the local models simultaneously learn both tool selection and argument generation. Fundamentally, these two capabilities differ from each other in that tool selection is a classification task (identifying appropriate tools from available options), while argument generation combines task-specific parameter selection with syntactic accuracy.

To overcome these constraints, we introduce a novel methodology that disaggregates the tool-calling task into two distinct specialized components: (1) **tool selection** and (2) **argument generation**. Leveraging this decomposition, we propose **decoupled fine-tuning**, a post-training approach that employs LoRA fine-tuning (Hu et al., 2022) to create dedicated adapters for tool selection and argument generation for individual tools, separately. Decoupled fine-tuning uses separate loss masking for tool selection and argument generation, and creates a dedicated LoRA adapter for each tool to generate its arguments, and a common LoRA adapter for the classification task of selecting the right tool at every step. We build an automated pipeline including synthetic data generation to produce diverse and high-quality training datasets and performing decoupled fine-tuning on the dataset to create the LoRA adapters.

This paper presents DualTune, an inference framework for local model orchestration that leverages the LoRA adapters generated through decoupled fine-tuning to perform efficient agent orchestration with the help of local models on end-user devices. DualTune breaks down every generation step into tool selection and argument generation, and dynamically loads the corresponding adapters for generating a tool call. Furthermore, DualTune implements *hierarchical orchestration* that enables it to support a large number of tools without compromising on accuracy. Hierarchical orchestration employs a second layer of decoupling that uses the base model to perform the high-level routing task of selecting the most appropriate toolset (such as `filesystem` or `notion`) at every step, which then routes the request to the tool selector that is trained on the particular toolset. This helps the tool selectors to choose from a restricted set of tools and thereby use a smaller context length containing only the restricted set of tools.

We use decoupled fine-tuning on the baseline Qwen-2.5-7B to create an LLM orchestrator called DualTuneModel-7B. Extensive experimentation across two benchmarks validates DualTune's effectiveness. Results demonstrate that DualTuneModel-7B significantly enhances tool orchestration quality compared to other local off-the-shelf and fine-tuned models of a similar size. DualTune also performs similar or better than local reasoning models, thus achieving high accuracy along with lower latency. Additionally, decoupled fine-tuning delivers $2\times$ higher improvement in tool calling compared to baseline on a standard MCP benchmark for the `filesystem` toolset, compared to traditional fine-tuning on the same dataset. DualTune operates efficiently on consumer-grade hardware, democratizing access to powerful and privacy-preserving agentic AI systems.

## 2 RELATED WORKS

### 2.1 LLMS ON END-USER DEVICES

Due to privacy concerns and the high costs of using advanced AI models through cloud APIs (Hasan et al., 2025; Hou et al., 2025; Guo et al., 2025), researchers have explored ways to run AI models

locally, including small foundation models (Belcak et al., 2025; Apple Inc., 2024) and systems support for improving efficiency (Frantar et al., 2022; Gerganov & ggml-org contributors, 2023).

## 2.2 LLMs as agent orchestrators

In recent years, there has been a significant interest in building autonomous agents based on LLMs that can execute complex tasks by calling an external tool. An increasing number of models, both in frontier models (OpenAI, 2025; Anthropic, 2025) and local models (Zhang et al., 2024; Lin et al., 2024; Liu et al., 2024b), support tool calling to function as agent orchestrators. The popularity of LLM agents has led to the creation of standard protocols, notably the Model Context Protocol (MCP) (Anthropic, 2024) MCP allows applications to expose themselves as MCP Servers, containing a list of tools along with their descriptions. Using this protocol, LLMs can connect to external applications and issue tool calls to the connected toolsets, allowing them to interact with the applications autonomously.

Additionally, there have been advancements in developing datasets, benchmarks, and leaderboards to assess the tool-calling capabilities of LLMs (Patil et al., 2025). In the context of MCP, benchmarks like MCP-Bench are designed to assess the capability of LLMs to generate MCP tool-calls (Wang et al., 2025), which involve realistic user prompts encompassing various applications and domains.

## 2.3 Post Training for Tool Calling

Some works propose post-training methods to improve tool calling capability. TinyAgent (Erdogan et al., 2024) performs fine-tuning with dataset generation that includes negative examples, and employs Retrieval Augmented Generation to select relevant tools corresponding to a user query, for reducing the context length of the model. Hammer (Lin et al., 2024) employs an augmented dataset that enhances models' sensitivity to irrelevant functions and incorporates function masking techniques to minimize misleading. ToolACE (Liu et al., 2024b) generates diverse tool-learning data to build a powerful local model.

## 3 Analysis

In this section, we analyze the effectiveness of local LLMs as orchestrators for agentic systems. To this end, we developed an agentic system in Rust that operates in a loop, beginning with a system prompt. It calls an LLM orchestrator to generate a tool call at each step. It then executes this tool in a containerized environment and appends the outcome to the conversational context, continuing to the next iteration. The workflow concludes when the orchestrator determines the task is complete and generates no further tool calls.

### 3.1 Evaluation Benchmark and Metric

| Model | ToolFit (%) | Reasoning |
|---|---|---|
| Qwen-2.5-7B | 16.0 | No |
| Qwen-3-8B | 52.3 | Yes |
| Qwen-3-8B* | 34.2 | No |
| Qwen-3-32B-Quant | 58.3 | Yes |
| Llama-3.1-8B | 42.4 | No |
| xLAM-2-8B | 15.8 | No |
| ToolAce-8B | 45.4 | No |
| GPT-5-mini | 88.4 | No |

Table 1: Performance of Local LLMs on MCP-Bench with the `filesystem` toolset

**Benchmark.** We use MCP-Bench (Wang et al., 2025), a recently proposed benchmark that evaluates LLM agents in realistic tool-use scenarios. Tasks in MCP-Bench are generated using an LLM-based synthesis pipeline which uses tool I/O signatures to create dependency chains of tool invocations, and then translates these dependency chains into natural language instructions, called "task descriptions". This is followed by a query-fuzzing phase where each task is rewritten such that it retains the core

objective but omits explicit tool references and execution steps. An example of the task description and the fuzzed query is in §C. We use MCP-Bench to create 50 tasks for the `filesystem` MCP toolset (Model Context Protocol, 2024), a collection of 12 tools for managing files and directories such as reading and writing files, creating directories, and querying metadata such as file size.

**Metric.** To ensure objective and scalable evaluation, we employ an LLM-as-judge methodology (Zheng et al., 2023). We use GPT-5 as the judge, which is provided with the ground-truth state of the filesystem for each task, allowing for a deterministic and accurate assessment of the final outcome. We assess orchestration performance using *ToolFit*, a metric which evaluates the output of agent systems on a 0-10 scale for tool-calling proficiency. The evaluation framework decomposes user queries and leverages the ground-truth data to establish an expected set of tool invocations that the agent system should produce. A model's score represents the proportion of expected tool outputs successfully generated relative to the expected tool calls. Note that there can be more than one tool that is suitable for a particular task; the judge is instructed to consider all such suitable tools as valid in §D. A perfect score of 10 indicates complete alignment, where the model correctly invokes all anticipated tools.

## 3.2 LOCAL MODELS EVALUATED

Based on the typical memory capacity of consumer-grade GPUs, we consider models smaller than 24GB in size for model weights as local models, so as to fit the model weights as well as the key-value cache in the GPU memory. We test a variety of these local models on the benchmark, as specified below:

**Off-the-shelf Local Models.** These include Qwen-2.5-7B (Yang et al., 2024), Llama-3.1-8B (Grattafiori et al., 2024), Qwen-3-8B (Yang et al., 2025) (with and without reasoning tokens) and Qwen-3-32B-Quant (Yang et al., 2025)

**Local Fine-tuned Models.** These include popular fine-tuned function-calling models, namely, xLAM-2-8B (Zhang et al., 2024) and ToolAce-8B (Liu et al., 2024b).

## 3.3 LOCAL MODELS ARE INCAPABLE AS LLM ORCHESTRATORS

As shown in Table 1, our evaluation reveals that existing local non-reasoning models perform significantly worse than the frontier GPT-5-mini model. The closest top-performing local model, Qwen-3-32B-Quant is a reasoning model, which incurs a high latency of generation, a known problem with the reasoning-based models (Liu et al., 2025). This shows that local models are not able to perform effectively as orchestrators for agentic systems.

**Prompt Tuning Is Not the Solution.** We next investigate whether a prompt-tuning-like approach can be used to significantly improve the performance of local models. While we cannot validate the effectiveness of prompt tuning exhaustively, we check the performance of Qwen-2.5-7B for the straightforward, non-fuzzed task descriptions (explained in §3.1) for the same 50 tasks. These tasks contain explicit instructions about tool names and tool arguments. We observe that even using explicit task descriptions incrementally improves the performance of Qwen-2.5-7B, from 16% to 22%. We show this impact of prompt tuning on other local models in §A.

| Tool Selection Model | Argument Generation Model | ToolFit (%) |
|---|---|---|
| Qwen-2.5-7B | Qwen-2.5-7B | 16.0 |
| Qwen-2.5-7B | GPT-5-mini | 28.8 |
| GPT-5-mini | Qwen-2.5-7B | 60.8 |
| GPT-5-mini | GPT-5-mini | 88.5 |

Table 2: Performance breakdown for Qwen-2.5-7B on MCP-Bench with the `filesystem` toolset

## 3.4 PROBLEMS IN TOOL SELECTION AND ARGUMENT GENERATION

To pinpoint the sources of failure in local models, we conducted an ablation study that decouples the two key stages of tool use: tool selection and argument generation. We achieve this by modifying the agent's workflow to use two separate LLM calls at each step: one to generate the tool's name and a second to generate its arguments.

This separation allows us to isolate each stage by substituting a frontier model's output for one of the steps. For example, to measure the quality of a local model's tool selection, we use it to choose the tool name but rely on GPT-5-mini to generate the arguments. Conversely, to assess its argument generation capability, we use GPT-5-mini to select the tool and the local model to generate the arguments. The results of this breakdown for the Qwen-2.5-7B local model are presented in Table 2.

We find that the performance bottleneck is primarily for the tool selection as changing the tool selector to GPT-5-mini improves the ToolFit from 16% to 60.8%. On the other hand, delegating the argument generation to GPT-5-mini only improves the performance from 16% to 28.8%. More importantly, this performance breakdown also suggests that tool selection and argument generation fundamentally differ in their nature, and can benefit from different optimizations. Fundamentally, tool selection is a classification problem, which requires identifying the appropriate tool at every step in the agent workflow from the available options. Argument generation, on the other hand, requires generating the right arguments as well as producing a structured output with syntactic accuracy.

### 3.5 Problems with Long Context due to Tool Descriptions

The attention mechanism of local models is often impacted due to long context inputs, as identified by prior work (Liu et al., 2024a; Qin et al., 2023). Even when the total number of tokens is smaller than the maximum context length that the model supports, a large context increases the complexity of the attention mechanism, causing the accuracy to drop. Furthermore, it also increases the latency of generation – the reasoning models (Qwen-3-8B, Qwen-3-32B-Quant) require 10-20$\times$ higher end-to-end time compared to the non-reasoning local models on the consumer-grade RTX 6000 GPU, for processing the queries in MCP-Bench

We run the same benchmark on Qwen-2.5-7B, but we increase the list of available tools to include tools from 2 other common applications: Notion and monday.com. This increases the total number of tools from 12 to 30. MCP mandates every tool to provide a description of its functionality, along with a json schema that contains the required and optional arguments for the tool. As a result of this, the tool descriptions account for a total of 9.8K tokens, as opposed to only 3K-4K tokens belonging to the actual task to be performed. We find that the increased context length reduces the ToolFit for Qwen-2.5-7B from 16% to 10.4%.

### 4 Efficient Local Agent Orchestration with DualTune

We introduce DualTune, a novel framework designed to enable the practical adoption and deployment of local LLMs in agentic systems via Low-Rank Adaptation (Hu et al., 2022). The core design principle of DualTune is **decoupled fine-tuning**, a strategy that decomposes the complex agent workflow at two levels.

At the first level, it separates tool selection from argument generation. This decoupling allows the creation of a dedicated LoRA adapters for the tool selection process and for the argument generation process. The output of decoupled fine-tuning is a single tool selection adapter and multiple argument generation adapters. The tool selection adapter is responsible for performing the classification task of choosing the right tool to invoke at every step, and every tool has a dedicated argument generation adapter that generates the tool-call arguments for its tool, given an input conversation history.

DualTune employs a second level of decoupling through *hierarchical orchestration*, which enables local models to operate on shorter context lengths, thereby benefiting both accuracy and latency. This strategy allows using a separate tool selector for every toolset (e.g., `filesystem` or `notion`), thus limiting the tool selector's decision space and improving its accuracy.

In this section, we detail the DualTune methodology. We begin by describing the fine-tuning process that includes our synthetic data generation strategy and the use of separate loss masking to train the tool selector and argument generator adapters. We then explain the hierarchical orchestration mechanism for managing a large number of tools. Finally, we discuss the integration of these components into a robust inference framework for effective local orchestration.

## 4.1 Decoupled Fine-Tuning Pipeline

The primary objective of the decoupled fine-tuning is to enhance the tool-calling accuracy of a general-purpose LLMs. This is achieved by fine-tuning two distinct types of LoRA adapters. First, we train a *Tool Selector* adapter that identifies the most appropriate tool among a given toolset to use at each step of a task. It effectively acts as a router, outputting only the name of the tool to be used. Second, we also train a separate *Argument Generator* adapter for each individual tool in the toolset. The sole responsibility of this adapter is to generate the correct and well-formed arguments required by its corresponding tool.

A critical requirement for this approach is a high-quality and diverse training dataset. To prevent the classifier from being biased towards frequently used tools and to ensure each argument adapter has sufficient data to learn its specific parameter schema, the dataset must contain a rich and evenly distributed set of examples for every available tool.

### 4.1.1 Synthetic Dataset Generation

To automate and scale the creation of our fine-tuning dataset, we leverage a powerful frontier model, GPT-5-mini, as a data generator. For each tool in a given toolset (*e.g.*, `filesystem`), we prompt GPT-5-mini to generate 1,000 diverse training examples, where each example consists of a user-like prompt that necessitates the use of that specific tool. The prompt used for this data synthesis process is detailed in the Appendix §B.

After generating the task prompts, we use the same frontier model (GPT-5-mini) to generate the tool-call sequences for each prompt[1]. We log the complete execution trajectories, which capture the initial prompt, the step-by-step LLM reasoning, the exact tool calls generated, and the resulting outputs from the tool. These high-quality trajectories serve as the ground truth for our fine-tuning process. While it may contain cases where GPT-5-mini performs poorly, that such cases are rare, given the high quality of the frontier model. After generation, we partition the entire set of trajectories into an 80% training set and a 20% validation set. In addition, we reserve 50 randomly sampled queries as a separate test set (also referred to as DualTune-TestSet which we evaluate in §5) that employs a balanced distribution to ensure coverage of all tools within each category. From each trajectory, we extract training instances for both the tool selection adapter and the relevant argument generation adapter.

### 4.1.2 Separate Fine-Tuning for Tool Selection and Argument Generation

The training of the tool selection and argument generation adapters requires different inputs and applies different loss masking during fine-tuning, since they focus on different parts of a tool call.

The tool selection adapter is trained on all the trajectories in the training set. The fine-tuning process involves predicting the name of the tool for the next step based on the context of all the prior steps and the user prompt for each trajectory. Since the aim of the tool selection adapter is only to choose the right tool (and not its arguments), we apply loss masking to compute loss only over the tool name tokens, not its arguments.

The argument generation adapter of a particular tool is trained on all the appearances of the tool within the training trajectories. The fine-tuning process involves predicting the arguments for the tool, given the context of all prior steps in the trajectory and the tool name for the current step. The loss is computed over the tokens that contain the arguments for the tool call.

## 4.2 Hierarchical Orchestration for Scalability

As the number of available tools grows, a single classifier must choose from an increasingly large set of options, which increases context length and the complexity of the tool selection task. To address this, we introduce hierarchical orchestration to manage multiple toolsets (*i.e.*, MCP servers).

This approach introduces a two-tiered tool selection process. At each step, we first use the *base* local model without any fine-tuned adapters to perform a high-level routing task: selecting the most

---

[1]Note that the use of cloud-based APIs does not pose privacy concerns in this context, as the training data can be generated within a controlled, dummy environment that excludes any sensitive information.

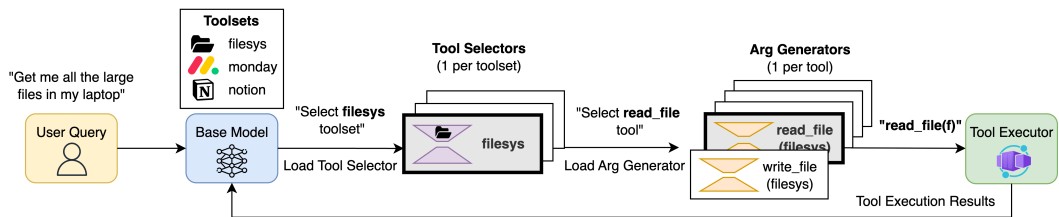

Figure 1: DualTune Inference Framework.

appropriate toolset (*e.g.*, `filesystem`). We find that this high-level routing is more straightforward than fine-grained tool selection and can be effectively guided by a system prompt and structured decoding, without requiring fine-tuning. Intuitively, this is because the tools belonging to different toolsets are vastly different since they belong to different applications. Selecting an appropriate toolset is thus a simpler task than selecting a tool from *within* a toolset.

Once a toolset is selected, we dynamically load the specialized tool selection adapter for that server. This tool selector then operates on a much smaller, more manageable set of tools (only the tools belonging to the selected toolset), significantly reducing the complexity of its decision. This hierarchical method effectively contains the context length and allows the system to scale to a larger number of tools without performance degradation. We evaluate hierarchical orchestration in §5.3.

### 4.3 INFERENCE FRAMEWORK

We integrate these techniques into the DualTune inference framework, a complete agentic orchestration framework implemented in Rust. For security and stability, DualTune executes all tools in containerized environments, providing fault isolation and resource management.

Our inference backend is powered by vLLM (Kwon et al., 2023), which is crucial for the practical feasibility of our approach. vLLM's ability to efficiently load, unload, and swap hundreds of LoRA adapters on the fly allows us to dynamically activate the necessary classifier and argument adapters at each step with minimal overhead. Fig. 1 shows the inference worflow in DualTune:

1. **Toolset Selection:** Given the current user prompt and conversation history, the base LLM selects the appropriate toolset (such as `filesystem` and `Notion`).
2. **Tool Selection:** DualTune loads the tool selection adapter for the selected MCP server. An inference call is made to this adapter, which returns the name of the desired tool.
3. **Argument Generation:** DualTune then dynamically swaps in the LoRA adapter specific to the chosen tool. A second inference call generates the precise arguments for the tool call.
4. **Execution and Observation:** The generated tool call is executed in a secure container. The output is captured and appended to the conversation history.
5. **Termination:** This loop continues until the tool selection adapter outputs a special token, "summarize". The system then generates a final summary for the user and terminates the execution.

### 5 EVALUATION

In this section, we present a comprehensive evaluation of different models and conduct ablation studies to analyze the contribution of individual components. We use DualTune to create an LLM orchestrator called DualTuneModel-7B, which creates the LoRA adapters for Qwen-2.5-7B.

### 5.1 EXPERIMENTAL SETUP

**Baselines.** We compare DualTuneModel-7B against local base models (Qwen-2.5-7B, Qwen-3-14B, Qwen-3-32B-Quant, Llama-3.1-3B), fine-tuned models (ToolAce-8B, xLAM-2-8B) and frontier models (GPT-OSS-20b, GPT-OSS-120b, GLM-4.5, Kimi-K2-Instruct, DeepSeek-v3.1).

**Tools Evaluated.** Our evaluation encompasses 30 MCP tools spanning three distinct toolsets (detailed below), representing common real-world applications. We conduct 50 queries per toolset for each benchmark to ensure statistical reliability.

| Model | DualTune-TestSet | | | MCP-Bench | | | Size | Reasoning |
|---|---|---|---|---|---|---|---|---|
| | filesys | monday | notion | filesys | monday | notion | | |
| **DualTuneModel-7B** | 66.4 | 55.8 | 87.2 | 61.5 | 43.2 | 71.8 | 18GB | No |
| GPT-OSS-20B | 25.8 | 28.4 | 24.0 | 2.0 | 31.6 | 72.2 | 13GB | Yes |
| Qwen-2.5-7B | 7.0 | 45.4 | 37.0 | 15.0 | 19.2 | 33.4 | 14GB | No |
| Llama-3.1-8B | 9.4 | 29.4 | 40.2 | 24.0 | 13.8 | 13.4 | 15GB | No |
| xLAM-2-8B | 2.6 | 40.2 | 43.0 | 15.4 | 18.2 | 12.8 | 15GB | No |
| ToolAce-8B | 14.0 | 36.6 | 14.0 | 45.4 | 7.4 | 3.4 | 15GB | No |
| Qwen-3-32B-Quant | 43.0 | 56.2 | 78.8 | 58.6 | 37.0 | 85.6 | 18GB | Yes |
| Qwen-3-14B | 21.6 | 45.0 | 68.8 | 62.6 | 29.2 | 79.6 | 28GB | Yes |
| GPT-OSS-120B | 40.4 | 62.0 | 52.2 | 46.6 | 55.0 | 84.2 | 66GB | Yes |
| GLM-4.5-Air-FP8 | 80.2 | 69.8 | 43.2 | 89.6 | 72.6 | 83.2 | 113GB | Yes |
| DeepSeek-V3.1 | 79.0 | 64.2 | 57.6 | 80.4 | 77.6 | 93.4 | 689GB | Yes |
| Kimi-K2-Instruct | 6.2 | 47.6 | 35.6 | 14.8 | 22.6 | 46.2 | 1.03TB | Yes |
| GPT-5-nano | 66.8 | 65.4 | 83.8 | 86.8 | 63.0 | 87.6 | / | Yes |
| GPT-5-mini | 67.4 | 60.2 | 87.0 | 88.4 | 76.4 | 91.6 | / | Yes |

Table 3: Performance of different base models on rena-bench and mcp-bench across three applications. The first 6 models we compare are local models that are smaller than 24GB in size.

- `Filesystem` toolset: This includes file and directory operations such as reading, writing, creating, and deleting files, as well as directory traversal and permission management tasks that form the backbone of system administration workflows.

- `monday.com` toolset: These tools encompass project management functionalities including task creation, status updates, team collaboration features, and workflow automation capabilities commonly used in enterprise project management environments.

- `Notion` toolset: This suite covers knowledge management operations including page creation, database queries, content organization, and collaborative editing features essential for modern documentation and information sharing workflows.

**Benchmarks.** We assess DualTune using two complementary benchmarks that evaluate different performance dimensions. The first benchmark is the test evaluation set of DualTune (called DualTune-TestSet), which includes 50 tasks for each toolset. The queries are designed to comprehensively test every tool in the toolset by generating roughly the same number of test cases for each tool, ensuring that all tools receive equal evaluation coverage. The second benchmark is MCP-Bench (Wang et al., 2025), an open-source benchmark featuring predefined system prompts that generate complex, fuzzed queries requiring coordination across multiple diverse tools, though with non-uniform tool usage patterns when compared to DualTune-TestSet as explained in §3.1. We do not evaluate queries with writes, since we cannot judge objectively based on the content generated by the LLMs for writes.

**Evaluation Metrics.** Model performance is assessed using ToolFit, a score ranging from 0 to 10 as detailed in §3.1. Our evaluation employs GPT-5 as the judge model, which analyzes trajectory lists alongside ground-truth data to compute ToolFit scores based on tool output quality and correctness.

## 5.2 OVERALL PERFORMANCE

Table 3 presents the comparative performance of all evaluated models across both benchmarks. We run all the models with the help of DualTune and enable hierarchical orchestration for all the models.

**DualTuneModel-7B versus local models without reasoning.** DualTuneModel-7B achieves a higher accuracy compared to all the other local models (Qwen-2.5-7B, xLAM-2-8B and ToolAce-8B) in both the benchmarks and across all the toolsets.

**DualTuneModel-7B versus reasoning models.** DualTuneModel-7B achieves a higher accuracy despite being a non-reasoning model compared to the local reasoning models (Qwen-3-14B, Qwen-3-32B-Quant) in 5 out of 6 cases.

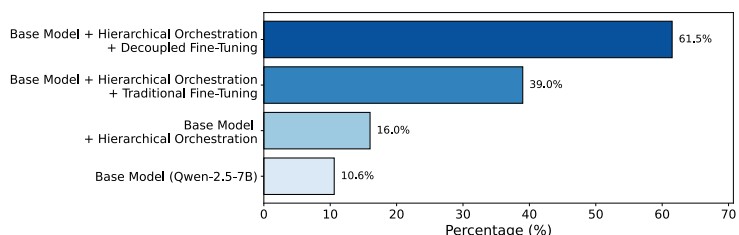

Figure 2: The contribution of each of the components of DualTune.

**DualTuneModel-7B versus base model.** The DualTune fine-tuning approach significantly improves the performance of DualTuneModel-7B compared to Qwen-2.5-7B on which it is post-trained, and achieves an accuracy up to 60% higher than Qwen-2.5-7B.

## 5.3 ABLATION STUDIES

To understand the individual contributions of DualTune's components, we conduct systematic ablation studies examining the effects of decoupled fine-tuning and hierarchical orchestration.

We first begin with the base model (Qwen-2.5-7B) without hierarchical orchestration. To this we add hierarchical orchestration that reduces the context length of the model due to using a restricted set of tools. Then we test the performance of Qwen-2.5-7B with hierarchical orchestration and traditional fine-tuning, creating a single LoRA adapter using the training data generated in DualTune. Finally, we evaluate DualTuneModel-7B, which is Qwen-2.5-7B with hierarchical orchestration and decoupled fine-tuning instead of traditional fine-tuning.

We perform this experiment on the `Filesystem` toolset for the queries generated using MCP-Bench The results for this ablation study are in Fig. 2. We find that hierarchical orchestration improves the accuracy of Qwen-2.5-7B from 10.4% to 16%. Using traditional fine-tuning further improves the accuracy from 16% to 39%. However, when we substitute traditional fine-tuning with decoupled fine-tuning for the same training dataset, we find that the accuracy jumps from 16% to 61.5%.

## 6 LIMITATIONS AND FUTURE WORK

DualTune relies on fine-tuning the base model to create a LoRA adapter to act as the tool selector for a toolset to select between available tools, and a separate LoRA adapter for each tool for argument generation. This has consequences in that adding new tools requires fine-tuning the common tool selector, as well as a separate LoRA adapter for the new tool. However, we believe that this is still a feasible approach because of two reasons. First, new tools are not added very frequently to existing tool sets. For example, there has only been one tool added to the `monday.com` toolset in the last five months, and no tools added to the `filesystem` toolset as well as the `Notion` toolset. This makes the fine-tuning process rare, and thus, practical. Second, we have a fully automated training pipeline as part of our contributions, which automatically generates the training dataset, performs the fine-tuning, and integrates the LoRA adapters as part of DualTune – the end-to-end process requires less than 10 hours in our experience, even while working with complex tool sets such as `monday.com` and `Notion`. In the future, we hope to explore techniques such as reinforcement learning to train the model while performing inference, thus automatically adapting to newer tools.

## 7 CONCLUSION

This paper proposes *decoupled fine-tuning*, a novel post-training approach that enhances the tool calling capability of LLMs by disaggregating a tool-call into tool selection and argument generation sub-tasks, and separately fine-tunes these sub-tasks. This paper presents DualTune, an LLM agent inference framework that uses LoRA adapters generated using decoupled fine-tuning to perform efficient agent orchestation on consumer-grade GPUs. We plan to open-source our fine-tuning pipeline (decoupled fine-tuning) and our inference framework (DualTune).

## 8 ETHICS STATEMENT

To the best of our knowledge, DualTune does not raise questions regarding the Code of Ethics.

## 9 REPRODUCIBILITY STATEMENT

The implementation of DualTune, along with instructions on how to set it up and reproduce the results in the paper, is uploaded to Supplementary Material.

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

APPENDICES

## A  IMPACT OF PROMPT TUNING

| Model | ToolFit (%) |
|---|---|
| Qwen-2.5-7B | 25.8 |
| Qwen-3-8B | 53.8 |
| Qwen-3-8B (No Reasoning) | 33.2 |
| xLAM-2-8B | 18.0 |
| ToolAce-8B | 25.6 |
| GPT-5-mini | 80.6 |

Table 4: Performance of Local LLMs on MCP-Bench using task descriptions instead of fuzzed queries on the MCP-Benchenchmark with the `filesystem` toolset

## B  PROMPT FOR QUERY GENERATION

The following passage shows the exact prompt used for the generation of file system queries:

**Prompt**

```
SYSTEM PROMPT: You are generating ONE English user request (exactly
one lines, no quotes, no code block, no numbering, no explanations).
GOAL: Create ONE requests that need to call $TOOL_NAME.
OUTPUT RULES (must follow all):
```

- Output EXACTLY TEN lines containing ONLY the user's request.
  - No prefixes/suffixes
  - No labels
  - No extra lines
- English only.
- Never include absolute paths.
  - Refer to location generically (e.g., "in the allowed directory",
    "within the permitted workspace", "inside the sandboxed area")
  - Or use a relative subpath (e.g., reports/2024-Q3.csv)
  - Do not include drive letters
  - Do not include leading "/" root paths
- Vary phrasing and structure aggressively:
  - Imperative / interrogative
  - Polite / terse / conditional
  - With or without "please"
  - Passive / active voice
  - Different synonyms for "allowed directory"
- Use diverse item names and extensions:
  - logs, csv, json, txt, md
  - pdf, png, jpg, mp3, mp4
  - zip, tar.gz, .env
  - Hidden dotfiles
  - Names with spaces/Unicode/UPPERCASE
  - Nested relative subfolders
- Include concrete numbers:
  - "first 10 lines", "last 5 lines"
  - "image1.png", "report.pdf"

```
    – "10 MB"
  • Do not mention any tool names, schemas, or parameters explicitly.
    The global list of tools is as follows:  $GLOBAL_TOOL_LIST
```

## C  AN EXAMPLE OF TASK DESCRIPTION AND FUZZED QUERY

**Task Description and Fuzzed Query**

**Task Description:**
1. Verify allowed directories.
2. Obtain a recursive directory tree for rena-browserd/browserd/src.
3. For exactly these four subdirectories, list entries with sizes
   and sort by size (sortBy=size):
    • rena-browserd/browserd/src/inference_engine
    • rena-browserd/browserd/src/container
    • rena-browserd/browserd/src/container_comm
    • rena-browserd/browserd/src/app_registry
4. From those four listings, identify the three individual files
   with the largest sizes that have a .rs extension.
5. For each of the top-3 files:
    • Read the first 50 lines (head=50).
    • Check whether a header is present, defined as:  at least two of
      the first ten lines start with // or begin a block comment /*.
6. Fetch file metadata for each top file:
    • Size
    • Last modified
    • Permissions
7. Produce a JSON report (array), one object per file containing:
    • path
    • size_bytes
    • header_present (true/false)
    • last_modified_iso
    • needs_deep_audit (true if size_bytes > 15000 **OR** last_modified
      is within the past 7 days)
    • Short recommendation string
  • Explicitly traverse only the four named subdirectories (no other
    directories).
  • Process exactly the three identified files (no more than three
    read_file/get_file_info calls each).
  • Treat relative dates as:  past 7 days and past 90 days when
    computing recency.
**Fuzzed Query:**  I'm trying to get a repo ready for a security
review for my project and my boss only wants me to look under
rena-browserd/browserd/src -- specifically those four subfolders
rena-browserd/browserd/src/container, rena-browserd/browserd/
src/app_registry, rena-browserd/browserd/src/inference_engine,
and rena-browserd/browserd/src/container_comm -- and I could
use help figuring out which three individual .rs files are the
largest by file size:  can you list the entries in each of those
four subfolders with sizes sorted by size so we can pick the top
three .rs files, then read the first 50 lines of each selected
file to check whether a header seems present (define header as
at least two of the first ten lines starting with // or /*),
fetch each file's size, last modified time, and permissions, and
return a JSON array (with paths given as relative paths from the
root permitted directory) where each object has path, size_bytes,

```
header_present (true/false), last_modified_iso, needs_deep_audit
(true if size_bytes > 15000 or last modified within the past 7
days), and a short recommendation ...
```

## D  PROMPT USED FOR COMPUTING TOOLFIT

**Prompt**

You are an impartial Judger that evaluates whether a tool-calling
trajectory adequately satisfied a filesystem-related query using
only the tools invoked and their raw outputs, not any assistant
summaries.
**Your job is to compute coverage**:
• Get the user query requirements from the query, for the output
• Map each user query requirement to tool requirements which can
  help achieve that requirement (every user query requirement must
  necessarily have one tool that is relevant to it).  This can be
  fetched by looking at tool descriptions
• Check if the tool output exists in the user trajectory item Score
  the trajector from 0-10 based on how many tool outputs exist, as
  a percentage of the required tool outputs
**Inputs**:You will be given four inputs:
• input_a (fs_status):  the ground-truth snapshot of the relevant
  filesystem (directory listings and/or metadata).
• input_b:  (tool descriptions):  This contains the descriptions of
  all the tools that were available for the trajectory.
• input_c (query):  the user's natural-language request (e.g.,
  list files, read file contents/data, fetch metadata like
  size/mtime/ctime/permissions, produce JSON, etc.).
• input_d (tool_call + tool_response):  the exact tools invoked and
  their raw outputs.  This is the only evidence of what the agent
  retrieved.  Assume there is no final assistant answer.
**Getting user query requirements**:
• Define the set of atomic requirements implied by the query:
  – Listing queries:  one atomic requirement per relevant item that
    should appear (file/dir).
  – Metadata queries:  one atomic requirement per (item,
    requested_field) pair (e.g., (fileA, mtime), (fileA, size)).
  – Content/data queries:  one atomic requirement per item whose
    contents are requested.  Treat as satisfied only if contents
    are shown and not truncated such that the task can be completed.
    If explicit ranges are requested (e.g., first 10 lines), treat
    each requested range as its own atomic requirement.
  – "All"/pattern scope:  expand the item set using fs_status (e.g.,
    all files under a directory or matching an extension/glob).
    Missing any item in scope yields missing atomic tool
    requirements for that item.
  – When uncertain about presence/completeness, or when elements
    are only implied (not evidenced in tool outputs), treat those
    atomic requirements as unsatisfied.
**Mapping user query requirements to tool requirements**:
• Based on the tool description and fs_status, map each user query
  requirement to the tool that can achieve it.  There MUST BE one /
  more tools that can help achieve EVERY user query requirement.
  If there are multiple tools that can achieve the user query
  requirements, then having any of them is valid.
**Your Tasks**
• List relevant files/directories.  From the query, derive a
  concrete list of relevant files/directories (explicit paths, or
```

those implied by patterns/globs).  If none, use [].  Prefer exact paths; include directories if applicable.

- Derive the tool requirements.  Using the user requirements, tool descriptions and fs_status, enumerate all atomic tool requirements as defined above.
- Evidence check (numerator).  Using only tool_response:
  - Mark an atomic requirement satisfied if the tool requirement item appears and the needed data for that atom is present (for metadata:  the specific field value; for content:  the actual content, not truncated; for listings:  the item name/path).
  - If content is clearly truncated (e.g., elided, "output too long"), mark the content requirement unsatisfied unless the query asked only for a subset that is fully present.
  - If fields are missing, wrong, or unverifiable, mark unsatisfied.
  - Extra/irrelevant outputs do not count against coverage; simply ignore them.
- Compute coverage.
  - coverage_percent = (satisfied_atomic_requirements / total_atomic_tool_requirements) * 100
  - If total_atomic_tool_requirements == 0, set coverage to 0% (cannot verify anything) and explain.
- Map to score (0–10).
  - Score_ToolCoverage = round(coverage_percent / 10) producing an integer from 0 to 10 (0%→0, 100%→10; 95%→10, 94%→9, etc.).  Use standard rounding (.5 rounds up).
- Output Format (strict JSON, no code fences, no extra keys)
  - Produce exactly:

    ```
    {
    "Reasoning_ToolCoverage": "<one concise paragraph that: (a) lists the relevant files/
    directories inline (e.g., [\"/path/a.txt\", \"/logs/b.csv\"]) or []; (b) summarizes the
    atomic tool requirements and how many were satisfied vs total; (c) states whether required
    contents/metadata/JSON values were fully evidenced or truncated/missing; (d) notes any
    missing scope items or fields that reduced coverage.>",
    "Score_ToolCoverage": <integer 0 to 10>
    }
    ```

  - Reasoning_ToolCoverage must be a single paragraph.
  - Score_ToolCoverage must be an integer between 0 and 10.

**Additional Principles**

- Judge only from tool outputs and fs_status; never infer unseen data or rely on assistant text.
- For "all"/pattern queries, completeness of scope is derived from fs_status.
- If any required element is unclear, unverifiable, or implied only by context, treat it as unsatisfied.
- Do not penalize formatting; coverage concerns presence and completeness of required data.

