# OpenReview forum: "DualTune: Decoupled Fine-tuning for On-Device Agentic Systems"
_ICLR.cc/2026/Conference — Submitted to ICLR 2026_

### Official Review · Reviewer_ws3Q · 2025-10-31

**Soundness:** 2
**Presentation:** 3
**Contribution:** 2
**Rating:** 2
**Confidence:** 3

**Summary:**

This paper presents a dualtune approach to fine tuning tool calling agents where instead of fine tuning the agent for the overall tool calling, they split the task into tool selection and argument filling and fine tune several LoRA adaptors (one for each function and tool selector) and show that this method can help with the on premise deployments to achieve better performance their base models and closes the gap with frontier models.

**Strengths:**

Show the detailed analysis of the approach using MCP bench and present various results compared to baseline models.

**Weaknesses:**

There are several other tool calling benchmarks and authors only presented numbers on a curated benchmark from their side and MCP bench. Results from other benchmarks would have been better to see.

I don’t understand the reason to train a new adaptor for each function for the argument-filling task. This seems like overkill. Did the authors try a single LoRA adaptor for all the functions and see if it can do the generic argument filling across tools?

Also, instead of LoRA adaptors, if I go with ICL examples for each function, will it give similar results as fine-tuning?

What are the inference time implications of LoRA adaptor switching? Any analysis around that?

Some of the basic baselines, like single argument filling model vs multiple, with respect to accuracy and time, comparison with simple ICL examples per selected function vs fine-tuned model needs to be shown to really present the case for the single function LoRA adaptor needed.

**Questions:**

Please check the weaknesses section.

---

### Official Review · Reviewer_5ja8 · 2025-11-01

**Soundness:** 2
**Presentation:** 2
**Contribution:** 2
**Rating:** 2
**Confidence:** 5

**Summary:**

Local LLMs struggle with tool selection and argument generation, so the authors split tool-calling into two subtasks and train separate LoRA adapters for each (“decoupled fine-tuning”). At inference, DualTune selects the tool, dynamically loads the matching adapter, and generates arguments with hierarchical orchestration to avoid unnecessary tools. On MCP-Bench, Qwen-2.5-7B with this method improves tool-calling accuracy by 46% and typically outperforms similar-size and even 2× larger local baselines.

The contribution of the paper is a decoupled fine-tuning method—instantiated as DualTuneModel-7B—that trains separate LoRA adapters for tool selection and argument generation. Across two benchmarks, DualTune outperforms similar-size local models, matches or beats local reasoning baselines at lower latency, and yields ~2× larger gains in tool-calling on an MCP filesystem benchmark than conventional fine-tuning. It also runs efficiently on consumer-grade hardware, enabling privacy-preserving on-device agents.

**Strengths:**

The strengths of the paper are,

- The method is simple and easy to understand, making it easy to follow.

**Weaknesses:**

The weaknesses of the paper can be summarized as,

- Introduction Part

    - “Accessible directly through LLMs.” Tools are accessed via the client/runtime that wraps the model, not “directly through LLMs.” This overstates the model’s capabilities and hides the execution layer.

    - Cost claim is overstated. “Simultaneously eliminating the costly API expenses associated with orchestration via frontier models.” Local inference removes per-token API fees but introduces compute, energy, engineering, and possibly remote tool/API costs. “Eliminating” is inaccurate.

    - Unsupported blanket claims. Phrases like “poor tool selection capabilities” and “poor argument generation capabilities” are strong generalizations about “existing local LLMs” without quantitative evidence (models, sizes, datasets, metrics, error bars).

    - Causal story is dubious. It claims large tool sets → “expanded context length … overwhelms attention mechanisms.” Long context alone doesn’t imply degraded attention for modern long-context models; failure is more often due to retrieval/noise or planning, not the mechanism being “overwhelmed.” Provide ablations isolating (a) number of tools, (b) prompt length, (c) retrieval quality.

    - Iterative repair claim lacks basis. “Limited ability to fix mistakes in subsequent steps” is not inherent to local models; it depends on the planner/executor loop and tool feedback. If this is empirical, report the loop design and repair rates.

    - Metric confusion. “Prompt tuning yield marginal accuracy improvements” — accuracy of what? Tool selection EM? End-task success? Provide concrete metrics and deltas; avoid vague performance statements.

    - Oversimplified task taxonomy. Framing selection as “a classification task” and parameter filling as “syntactic accuracy” is reductive. Selection often requires multi-label reasoning with constraints; parameter filling requires semantic grounding and schema validity, not just syntax.

    - Rhetorical emphasis over analysis. Multiple assertions (e.g., “not substantial”) are qualitative without confidence intervals or statistical tests. Academic writing should ground these claims in data and controlled comparisons.

- Method Part

    - Metric inconsistency. The text defines ToolFit as a 0–10 score, yet Table 1 reports percentages (e.g., 58.3, 88.4) and labels the column “ToolFit (%)”. These cannot both be true; specify one scale and keep it consistent.

    - LLM-as-judge overclaim. Calling the judge “deterministic and accurate” is unjustified. Even with temperature 0, LLM judgments can vary across prompts/versions; accuracy requires validation (e.g., agreement with human annotators), which is not provided.

    - The meaning of Qwen-3-8B* in Table 1?

    - Misleading “consumer-grade GPU” premise. Framing 24 GB as the “typical” consumer GPU capacity is inaccurate (8–16 GB is far more common). Conclusions based on this premise may not generalize.

    - Overgeneralized claim of incapability. From one toolset and 50 tasks the authors conclude “local models are not able to perform effectively as orchestrators.” That’s too strong: it ignores other toolsets, planners, retrieval strategies, and stronger local models; also the baseline GPT-5-mini is non-public, making the comparison unverifiable.

    - Latency claim without evidence. “Reasoning models incur high latency … a known problem” is asserted without measurements (decode lengths, tokens/sec) or citations that actually establish this for the authors' setups.

    - Confounded “decoupling” ablation. Swapping in GPT-5-mini to generate arguments when evaluating a local model’s tool selection (and vice-versa) does not isolate a single stage: ToolFit depends on both the chosen tool and the validity/format of its arguments. Using a much stronger model for the other stage changes the distribution and can inflate the measured gain, so attributing the 16%→60.8% jump “primarily to tool selection” is not warranted without a control that keeps the counterpart stage constant and equivalently capable.

    - Causal claim about long context is overgeneralized. “Large context increases the complexity of the attention mechanism, causing the accuracy to drop” is asserted as general fact with citations but no controlled evidence from the authors' setup (no ablation of prompt length vs. retrieval/noise, no per-token loss/attention diagnostics). It may be true in some cases, but the blanket causality is unsupported here.

    - Attribution error in the long-context experiment. The authors change two variables at once—tool count (12→30) and description length (≈3–4k→9.8k tokens)—then ascribe the drop (16%→10.4%) to context length. Increased choice set size alone can reduce ToolFit. A valid test must hold tool count constant while varying description length (or vice versa).

    - “Ground truth” misuse and unsupported quality claim. LLM-generated trajectories are labeled as ground truth and asserted to be “high-quality” with “rare” errors, but no human validation rates or QC protocol are reported.

    - Circularity/contamination risk. The same teacher model generates both the training data and the separate “DualTune-TestSet,” and earlier sections also use the teacher as a judge. This teacher-generated test plus teacher-as-judge setup can inflate performance and undermines independent evaluation.

    - Scalability mismatch. Training a separate argument-generator adapter per tool scales linearly with tool count and contradicts the claim of a “general-purpose” approach; memory/maintenance costs are not addressed.

    - Test set design is weak/ambiguous. A fixed 50-query test set across all tools is too small for reliable estimates and appears to be sampled from the same distribution as training prompts, with unclear disjointness from the 80/20 split.

    - Unsubstantiated scaling claim. “Allows the system to scale to a larger number of tools without performance degradation” is too strong. Hierarchical routing adds extra LLM calls and dynamic adapter loads; any savings from shorter prompts must be shown to outweigh these costs with measurements.

    - Method underspecified / hard to reproduce. Phrases like “guided by a system prompt and structured decoding” lack operational detail (inputs seen at tier-1, decoding rules, thresholds). Without this, others cannot reproduce or evaluate the claimed benefit.

   - Token/complexity accounting is missing. The authors argue context length drives difficulty but don’t quantify how much text the tier-1 router reads (names vs. descriptions) nor the end-to-end token, memory, and latency budget versus the flat baseline.

- Experiments Part

   - Math/number inconsistencies. The baseline is 10.6% in Fig. 2 but 10.4% in the text below. You also write “up to 60% higher than Qwen-2.5-7B,” yet 61.5 vs 16.0 is +45.5 percentage points (≈+184% relative), and 61.5 vs 10.6 is +50.9 pp (≈+480%). Say “percentage-point gain” and ensure the baselines match.

    - Unfair ablation setup. You “perform this experiment on the Filesystem toolset,” yet claim hierarchical orchestration helps by restricting to the correct toolset. If evaluation only contains filesystem tasks, letting the hierarchical method filter to the filesystem toolset effectively gives it oracle prior knowledge and shrinks the search space, while the flat baseline appears to consider all tools. Make the candidate tool sets identical for all variants or route on a mixed multi-toolset workload.

    - Component attribution is confounded. The jump 16%→39% (traditional FT) and 16%→61.5% (decoupled FT) use training data generated by your own pipeline; without a held-out, independently sourced test set and variance/error bars, the “contribution of each component” plot is not trustworthy.

    - Reproducibility gap. The model name “DualTuneModel-7B” and training details (LoRA ranks, data volume per tool, selection prompts) are insufficient to reproduce these exact numbers; the figure has no confidence intervals or multiple-seed runs.

**Questions:**

Please refer to the Weaknesses section for details. Overall, the manuscript lacks sufficient novelty and technical depth for this area. The problem motivation and positioning are unclear; the method relies on several unvalidated assumptions; experiments are limited in scope, with narrow model/data choices, and the reported results do not adequately support the claims. Descriptions of implementation and workload are insufficient, hindering reproducibility. I recommend clarifying the contribution boundaries, adding strong baselines and ablations, and providing fuller methodological details before resubmission.

---

### Official Review · Reviewer_K4SY · 2025-11-01

**Soundness:** 2
**Presentation:** 2
**Contribution:** 2
**Rating:** 2
**Confidence:** 4

**Summary:**

This paper proposes a decoupled strategy for tool use, which divides the tool-using process into two stages: tool selection and argument generation. For each toolset, the authors train an independent LoRA module to enhance tool-specific performance. Experiments on the MCP-Bench dataset are conducted to demonstrate the effectiveness of the proposed approach. While the idea of decoupling tool-related sub-tasks is interesting and potentially beneficial, the paper overlooks important aspects regarding efficiency, scalability, and inter-toolset interaction, which limit the practicality of the approach.

**Strengths:**

1. The paper presents a motivated and intuitive decoupling idea, which offers a new perspective on tool-use modeling.
2. The writing is clear and the framework design is easy to follow, making the overall contribution accessible to readers.

**Weaknesses:**

1. The paper should provide a clearer illustration of the statistics and composition of the evaluation benchmark (e.g., number of tools). This would help readers better understand the experimental setup and the claimed generalization ability.
2. The proposed approach introduces two sequential inference processes (tool selection and argument generation), which substantially increases computational cost. Since these are performed using separate models, KV caching and similar acceleration techniques cannot be shared between stages, leading to high latency and memory overhead. The problem is further exacerbated in multi-turn function-calling scenarios, where model switching and repeated prefill computation may result in severe performance degradation. An analysis of training/inference efficiency or latency comparison with unified models would make the work more convincing.
3. The current framework assumes independence across toolsets, which restricts its applicability to tasks involving inter-tool dependencies (e.g., when the output of a tool in one set serves as input for another). Such relationships are common in complex workflows, but the proposed model cannot capture them. Moreover, the inference cost may grow linearly with the number of toolsets, making it impractical for large-scale deployment.
4. More comprehensive ablation studies are needed to justify the decoupling strategy. For instance, combining a well-trained tool selector with a base model as the argument generator to isolate the effect of each component.

**Questions:**

Please refer to the weaknesses

---

### Official Review · Reviewer_aZ6j · 2025-11-01

**Soundness:** 2
**Presentation:** 3
**Contribution:** 2
**Rating:** 4
**Confidence:** 4

**Summary:**

This paper addresses the performance shortcomings of local LLMs deployed in on-device agentic systems. It reveals the suboptimal performance of local LLMs in tool orchestration through detailed analysis. To address these issues, the authors propose a "decoupled fine-tuning" method and the "DualTune" inference framework, whose main idea is to decompose the tool-calling pipeline into tool selection and argument generation. Furthermore, the paper argues for the importance of toolset separation, which can effectively boost the performance of tool selection compared to direct fine-grained tool selection. Extensive experimental results demonstrate the effectiveness of the proposed tuning method and inference framework.

**Strengths:**

1. The analysis section is comprehensive and convincing. It also serves as a strong foundation for the proposed method and inference framework.
2. I agree with the finding that selection among fine-grained tools leads to overly long contexts, which further degrades the tool-selection capabilities of LLMs. To resolve this, the proposed 2-tiered tool-selection process is reasonable and efficient.
3. The experimental results demonstrate the effectiveness of the proposed methods.
4. The writing and structure of the paper are clear and easy to follow.

**Weaknesses:**

1. As illustrated in the Limitations section, the scalability of the proposed method is limited, as it requires further fine-tuning to adapt to new tools.
2. The experimental setup is somewhat unfair. In the main experiment, the DualTune-testset is an in-domain test set (since DualTuneModel-7B is trained on the corresponding training set), and tools in MCP-Bench are also fine-tuned using synthetic data. It would be more convincing to demonstrate the effectiveness of tool-selection/argument-generation separation and the two-tiered selection if the authors evaluated on some OOD benchmarks or toolsets.
3. In Line 294, the authors state that "while it may contain cases where GPT-5-mini performs poorly, such cases are rare." It would be more convincing if the authors provided quantitative results.
4. The formatting of the paper title does not meet the ICLR 2026 guidelines.

**Questions:**

1. Can you provide some examples of GPT-5 generated data?
2. What is the average additional time required for DualTuneModel-7B to complete a full cycle of “toolset selection -> tool selection -> argument generation -> tool execution” compared to direct tool calling?

---

### Meta-Review · Area_Chair_Tyry · 2025-12-14

**Summary:**

While the analysis grounds the motivation for decoupling tool selection from argument generation, the in-domain nature of the DualTune test set raises concerns about the robustness of the results. Additionally, the reliance on synthetic fine-tuning data for MCP-Bench may limit the generalizability of the conclusions. The method's scalability is also constrained, as it requires additional fine-tuning for each newly introduced tool. Moreover, several claims in the paper lack sufficient support from controlled ablations or quantitative analysis.

**Reviewer Concerns:**

No author response was submitted.

**Reviewer Scores:**

No author response was submitted.

---

### Decision · Program_Chairs · 2026-01-26

Reject